# The Association between Menstrual Irregularities and the Risk of Diabetes in Premenopausal and Postmenopausal Women: A Cross-Sectional Study of a Nationally Representative Sample

**DOI:** 10.3390/healthcare10040649

**Published:** 2022-03-30

**Authors:** Byung-Soo Kwan, Seung-Chan Kim, Hyen-Chul Jo, Jong-Chul Baek, Ji-Eun Park

**Affiliations:** 1Division of Gastroenterology and Hepatology, Department of Internal Medicine, Samsung Changwon Hospital, School of Medicine, Sungkyunkwan University, Changwon-si 51353, Korea; byungsoo2459.kwan@samsung.com; 2Biostatistics Cooperation Center, Gyeongsang National University Hospital, Jinju-si 52727, Korea; seungchan.statistics@gmail.com; 3College of Medicine, Gyeongsang National University, Jinju-si 52828, Korea; 73hccho@gnuh.co.kr (H.-C.J.); gmfather@gnu.ac.kr (J.-C.B.); 4Department of Obstetrics and Gynecology, Gyeongsang National University Changwon Hospital, Changwon-si 51472, Korea

**Keywords:** diabetes, menopause, menstrual irregularities

## Abstract

Background: Studies have assessed the effects of menstrual irregularities and menopause on diabetes, but no definitive conclusion has been reached. This study investigated for the first time the relationship between menstrual irregularity and diabetes before and after menopause. Methods: This population-based cross-sectional study included 9043 participants from the Korea National Health and Nutrition Examination Survey (KNHANES) V (2010–2012). Multivariate logistic regression was used to assess the effect of menstrual irregularities on impaired fasting glucose (IFG) and diabetes incidence in women before and after menopause. Results: After adjustment for age and other diabetes-related factors, both menopause (OR = 1.51, 95% CI = 1.101–2.27, *p* = 0.047) and menstrual irregularities (OR = 1.51, 95% CI = 1.1–2.07, *p* = 0.011) were found to increase the risk of diabetes. Menstrual irregularities were significantly related to diabetes in the postmenopausal group (OR = 1.65, 95% CI  =  1.12–2.42, *p* = 0.012) but not in the premenopausal group (OR = 1.22, 95% CI  =  0.64–2.32, *p* = 0.555). Conclusions: In this study, menopausal status appeared to independently affect diabetes risk; menstrual irregularities were found to be a risk factor for postmenopausal diabetes. This study emphasizes the need for monitoring and early prevention, along with medical advice on menstrual irregularities, to reduce the prevalence of diabetes and improve the quality of life of postmenopausal women.

## 1. Introduction

Diabetes is a widespread public health issue with increasing prevalence, and causes microvascular and macrovascular diseases [1]. According to the 2017 Global Disease Control Study (GBD), approximately 462 million people are affected by diabetes, accounting for 6.28% of the world’s population [2]; the prevalence of diabetes worldwide is expected to increase to 552 million by 2030 [3]. Accordingly, efforts to understand the risk factors for diabetes, along with public health and clinical precautionary measures designed to lower the risk of diabetes, are important. One of the risk factors for diabetes is obesity [4], and menopause is expected to be associated with diabetes in women, as it is accompanied by changes in body composition, glucose metabolism, and insulin sensitivity [5].

Menstrual irregularities are considered an important indicator of women’s health. In particular, they are closely related to pathological conditions, including polycystic ovarian syndrome (PCOS), and to reproductive dysfunction and metabolic imbalance. PCOS has a prevalence of 4.8–8% in the general population [6], and the prevalence of menstrual irregularities is reported to be 11–27% [7,8,9]. However, the effects of PCOS and menstrual irregularities on individuals and healthcare providers remain underestimated.

The association between irregular menstrual cycles and various diseases has been demonstrated previously [10,11,12]. Highly irregular menstrual cycles are associated with an increased risk of rheumatoid arthritis, breast cancer, cardiovascular disease, metabolic disease, and diabetes. Several studies have reported the association between diabetes and menstrual irregularities. An investigation involving nurses in their 20s and 40s in the United States reported that menstrual irregularities increased the risk of developing type 2 diabetes [12,13], and another report indicated that menstrual irregularities in adolescents younger than 19 years of age were associated with type 1 diabetes [14]. On the other hand, a prospective study of women in their 20s and 30s found that there was no association between menstrual irregularities and diabetes among adults [15]. Furthermore, research on women in their 50s and 70s also revealed that previous menstrual irregularities were not associated with type 2 diabetes [16]. As such, the correlation between menstrual irregularities and diabetes has been inconsistent depending on the subjects involved. No studies have investigated the effect of menstrual irregularities on diabetes before and after menopause, which is considered to be related to the onset of diabetes.

The aim of this study was to investigate whether menstrual irregularities and menopause are related to diabetes risk, and to identify the association between menstrual irregularities and impaired fasting glucose (IFG) or diabetes in women before and after menopause, using nationally representative data.

## 2. Materials and Methods

### 2.1. Study Participants and Design

This study employed a cross-sectional design. All data are available in the Korea National Health and Nutrition Examination Survey (KNHANES) database (http://knhanes.cdc.go.kr (accessed on 8 December 2021)). This study was based on 3-year data (2010–2012) obtained from the KNHANES V. The KNHANES is a representative survey established in 1998 that periodically evaluates the health and nutritional status of Korean individuals and investigates health risk factors. Approximately 10,000 new samples are collected every year. Participants are selected using proportional allocation systematic sampling with multistage stratification.

A total of 25,534 persons participated in the KNHANES 2010–2012. The participants included 11,016 women over the age of 20, of whom 9143 had provided information on menstrual irregularities and diabetes risk. We therefore analyzed 9043 women over 20 years old, with complete data on reproductive factors and clinical variables (Figure 1). Each participant provided written informed consent.

Participants were divided into normal, IFG, and diabetes groups according to their diabetes status, and each variable was compared among these groups.

In addition, we compared these variables between premenopausal and postmenopausal women in the diabetes status groups.

In the premenopausal and postmenopausal groups, the association between menstrual irregularities and IGT or diabetes was determined using multivariate logistic regression analysis.

### 2.2. Study Variables

#### 2.2.1. Menstrual Irregularity

In the health survey, the participants were asked, “Is your menstruation regular?” Responses were either “yes” (regular menstrual cycles) or “no” (missing three or more periods in a row). The response was dichotomized for subsequent analysis. “No” responses were taken to indicate menstrual irregularities.

#### 2.2.2. Menopause

Premenopausal women were defined as having no changes in menstrual cycles, and postmenopausal women were defined as having experienced a permanent discontinuation of menstruation for at least 12 consecutive months.

#### 2.2.3. Diabetes Status

This analysis focused on the risks of IFG and diabetes as outcomes. Participants were divided into three subgroups according to their diabetes status: the normal, IFG, and diabetes subgroups. Diabetes was defined as a previous diagnosis of diabetes or the use of medication for diabetes based on self-reports or the survey results of fasting plasma glucose (FPG) levels ≥126 mg/dL (7.0 mmol/L) and/or an HbA1c level ≥6.5%. Impaired fasting glucose was defined as an FPG level of 100–125 mg/dL (5.6–6.9 mmol/L) and an HbA1c level <6.5%. Normal glucose was defined as an FPG level <100 mg/dL (5.6 mmol/L). All participants fasted for at least 8 h before blood samples were collected. Plasma glucose was directly measured using a Hitachi Automatic Analyzer 7600 (Hitachi, Tokyo, Japan)

#### 2.2.4. Covariates

We collected data on demographic and health-related variables, including age, waist circumference (WC), body mass index (BMI), smoking status, blood pressure, type of exercise, number of pregnancies and history of oral contraceptive use.

Trained medical staff performed anthropometric measurements following a standardized procedure. Waist circumference was measured to the nearest 0.1 cm at the narrowest point between the lower border of the rib cage and iliac crest after normal expiration. Height and body weight were measured to the nearest 0.1 cm and 0.1 kg, respectively, with the participants wearing light clothing after removing their shoes. BMI was calculated as weight (kg) divided by the square of height (m^2^), and participants were subsequently categorized as underweight (<18.5 kg/m^2^), normal weight (≥18.5 and < 25 kg/m^2^), overweight (≥25 and < 30 kg/m^2^), or obese (≥30 kg/m^2^). Blood pressure was measured 3 times by trained nurses using a mercury sphygmomanometer (Baumanometer; Baum Co., Copiague, NY, USA); during measurements, participants were in a seated position with the arm supported at heart level after 5 min of rest. Patients were divided into 3 groups according to their hypertensive status. Hypertension was defined as systolic blood pressure (SBP) ≥ 140 mmHg, diastolic blood pressure (DBP) ≥ 90 mmHg or the use of an antihypertensive medication. Prehypertension was defined as 120 mmHg ≤ SBP < 140 mmHg or 80 mmHg ≤ DBP < 90 mmHg. Normal blood pressure was defined as SBP < 120 mmHg and DBP < 80 mmHg.

Demographic and health-related variables were obtained from self-report questionnaires and personal interviews conducted by trained staff. Participants were categorized as current smokers or nonsmokers, regardless of past smoking history. Types of exercise included aerobic and muscle exercises. Aerobic exercise was defined as exercise that met the standards for aerobic activity recommended by the World Health Organization (WHO) guidelines [17], and muscle exercise was defined as strength-training exercise at least twice a week. The total number of pregnancies was provided as an integer. Participants were also categorized into two groups based on oral contraceptive use: OC use for ≥1 month during a lifetime or not.

### 2.3. Statistical Analysis

To integrate the 3 years of data, sampling weights were applied, and one data point was produced. Differences in the variable distributions by the three glucose status groups were assessed by Rao–Scott X2 tests (function svychisq in the R package survey) for categorical variables and linear regression models (function svyglm in the R package survey) for continuous variables. To analyze the risk of impaired fasting glucose and diabetes with reference to the normal controls, univariate analysis was performed for each investigated variable. Then, multivariate logistic regression analysis was performed to evaluate the risk of impaired fasting glucose and diabetes according to menstrual irregularities. Variables with statistically significant associations in the univariate analysis were included in the multivariate logistic regression analysis. We adjusted for 7 confounder variables (age, smoking status, BMI, hypertension, number of pregnancies, oral contraceptive use, and exercise) in the regression analysis. In addition, the risk of impaired-fasting blood glucose and diabetes according to menstrual irregularities was also analyzed. All analyses were conducted using R software version 4.0.3 (R Core Team, R Foundation for Statistical Computing, Vienna, Austria, 2020). For all analyses, *p* values < 0.05 were considered indicative of statistical significance.

### 2.4. Ethics

The data from the KNHANES were anonymized prior to release to the public from the Korea Centers for Disease Control and Prevention. This study was approved by the Korea Centers for Disease Control and Prevention Institutional Review Board (IRB Nos. 2010-02CON-21-C, 2011-02CON-06-C, and 2012-01EXP-01-2C).

## 3. Results

The mean age of the participants (N = 9043) was 46.6 ± 15.8 years. Menstrual irregularities were experienced by 1003 (12.5%) participants, and 4638 (40.7%) participants were postmenopausal. The baseline characteristics of the study participants, grouped by diabetes status, are shown in Table 1. In total, 811 (9%) participants had diabetes, and 1436 (15.9%) had impaired fasting glucose. In addition to menstrual irregularities, our analysis included the following baseline characteristics: age, waist circumference (WC), body mass index (BMI), smoking status, blood pressure status, exercise level, menopause, number of pregnancies, and oral contraceptive use. Except for menopause and smoking status, all variables showed significant differences across the three diabetes status groups, including significant differences in menstrual irregularities.

In addition, Table 2 shows the variable values when the diabetes status groups were subgrouped according to menopausal status (premenopausal or postmenopausal). There were no significant differences in smoking status, menstrual irregularities, or oral contraceptive use between these subgroups, but other variables showed significant differences depending on menopausal status.

We estimated the risk of IFG and diabetes according to menstrual irregularities and menopausal status using an adjusted odds ratio (Table 3). Overall, the risk of diabetes was higher in the postmenopausal group (adjusted OR = 1.51, 95% CI = 1.01–2.27, *p* = 0.047) than in the premenopausal group. The risk of diabetes was also higher in the menstrual-irregularity group (adjusted OR = 1.51, 95% CI = 1.1–2.07, *p* = 0.011) than in the regular menstrual cycle group. Within the postmenopausal group, the risk of diabetes was higher for women with menstrual irregularities (adjusted OR = 1.62, 95% CI = 1.12–2.42, *p* = 0.012) than for women with regular menstrual cycles. Within the premenopausal group, menstrual irregularities were not significantly associated with the risk of diabetes (adjusted OR = 1.22, 95% CI = 0.64–2.32, *p* = 0.555). There was no significant association between the occurrence of IFG and menopausal status or menstrual irregularity.

## 4. Discussion

We reported an association between menstrual irregularities and diabetes among adult women before and after menopause based on data from the KNHANES V database (from 2012 to 2015). Our main results are that after adjustment for factors such as age and obesity, postmenopausal women had a 51% higher risk of diabetes than premenopausal women, and that for postmenopausal women, previous menstrual irregularities increased the risk of developing diabetes by 65%. To our knowledge, this is the first investigation to examine the relationship between menstrual irregularities and diabetes before and after menopause.

Our results showed that 12.5% of adult women had experienced or were currently experiencing menstrual irregularities. According to the literature, the prevalence of menstrual irregularities varies from 5% to 27% depending on age, occupation, and ethnicity [7,8,9]. This variability may be due to the different criteria for menstrual irregularities used by researchers and the different age distributions of participants. Additionally, data about menstrual irregularities are obtained only from self-reports.

Whether or not menopause is related to the development of diabetes is controversial. A prospective study involving 8099 participants reported that menopause did not increase the risk of diabetes after adjustments for age, BMI, waist circumference, and smoking status, which were found to be risk factors for diabetes [18]. In contrast, the odds ratios (ORs) of type 2 diabetes and prediabetes were found to increase significantly among postmenopausal women, regardless of age, compared to those of premenopausal women [19]; moreover, surgical menopause [20] and early menopause [16,21] have been reported to increase the risk of diabetes. Our results indicated that, after adjustments for age, BMI, smoking status, high blood pressure, and physical activity—which are risk factors for diabetes—postmenopausal women had a 51% higher risk of diabetes than premenopausal women. Possible explanations for the inconsistent outcome are that the assessment of menopausal status is usually based on self-reported responses or interviews, and that participant race and characteristics vary across studies. Additionally, our data did not distinguish between natural and surgical menopause. We also lacked oral glucose tolerance test (OGTT) data, which are typically used to diagnose diabetes.

We found that women who reported irregular menstrual cycles had an increased risk of developing diabetes. This is in line with previous research that observed an association between menstrual irregularities and the development of diabetes [12,13,22,23]. Previous studies have found that irregular menstrual cycles were associated with a higher risk of insulin resistance [24] and gestational diabetes [25]. However, other studies have reported no association between irregular menstrual cycles and the risk of type 2 diabetes [15,16]. Notably, one of these studies [16] examined menopause only between the ages of 50 and 79. As the authors noted, an increased risk of diabetes occurs earlier in women with PCOS (in their 30s and 40s) than in the general population (60s and 70s). The authors assumed that there was no correlation between menstrual irregularities and diabetes because of the age of the participants in their study. However, the fact that PCOS—which is characterized by irregular and long menstrual cycles—is explained by impaired glucose tolerance, insulin resistance, and obesity, supports our results and the preponderance of evidence.

As mentioned above, debates persist as to whether menopause independently influences the development of diabetes [18,19]. However, the association between menopause and changes in body composition related to adverse insulin sensitivity and glucose metabolism [26,27], as well as the finding that hormone therapy helps control glucose levels in postmenopausal women diagnosed with diabetes [28], suggest that there is an association between menopause and diabetes. Thus, although we did not find a significant association between menstrual irregularities and premenopausal diabetes, there was a significant association of menstrual irregularities with postmenopausal diabetes. A possible reason for this result is that the contribution of factors influencing the development of diabetes may differ between perimenopausal and postmenopausal women [29,30]. It is postulated that the long-term effects of estrogen on insulin secretion and glucose homeostasis [31] may have contributed to these results.

The main strengths of this study are that it is based on a nationally representative population and has a large sample size. In previous reports [12,13,15,16,22,23], the association between menstrual irregularities and diabetes appeared inconsistent depending on the age and menopausal status of the sample group. In this study, the group was divided into premenopausal women and postmenopausal women. In addition, the results of our multivariate analysis were adjusted for potential factors that influence diabetes.

This study has some limitations. Firstly, because of the cross-sectional design, the causal relationships of menstruation irregularities and menopausal status with diabetes could not be determined. Secondly, data on menstrual characteristics were limited and obtained from individuals’ responses to one question about menstrual regularity, which could be influenced by potential recall bias and misclassification. In addition, misclassification could have occurred because menopause was defined as a lack of menstruation for more than 12 months without specific endocrine parameters related to the diagnosis of menopause, the differences in symptoms or duration [32], and/or because the information used to categorize subjects according to menopausal status was obtained only from self-reported information on the questionnaire. In other words, it is not clear whether the cause of irregular menstrual cycles for women in their 40s is related to menopausal transition or an enduring irregular menstrual pattern. Data from tests such as the OGTT, which are used to assess beta-cell function and insulin resistance [33], were not included, and with the available information, we could not identify a family history of diabetes or a history of gestational diabetes, which are closely related to the development of diabetes. Although type 1 diabetes and type 2 diabetes have different etiologies [34], we could not distinguish the type of diabetes in this study.

Despite these limitations, our study’s strengths are that it analyzed data from a large, nationally representative sample, and adjusted for confounding factors related to diabetes. Our findings highlight the importance of screening for diabetes in individuals with irregular menstrual cycles, and we are confident that they will facilitate the establishment of policies to reduce the risk of developing diabetes in women with irregular menstrual cycles.

## 5. Conclusions

This nationally representative, population-based cross-sectional study found that previous menstrual irregularities were highly associated with the occurrence of diabetes in women, and that the risk was higher among postmenopausal women. In addition, menopause itself was related to the occurrence of diabetes in women even after adjustment for age and other risk factors for diabetes. However, since this is a cross-sectional study involving a survey, future studies are needed to confirm our observations and/or extend our findings. Well-designed prospective long-term follow-up studies or mechanistic studies are needed to clarify the causal relationship; based on their findings, stronger suggestions and guidelines can be presented. Nevertheless, our findings support the need for ongoing diabetes-related monitoring and screening among women with a history of menstrual irregularities, along with medical advice on menstrual irregularities, to reduce the prevalence of diabetes and improve the quality of life of postmenopausal women.

## Figures and Tables

**Figure 1 healthcare-10-00649-f001:**
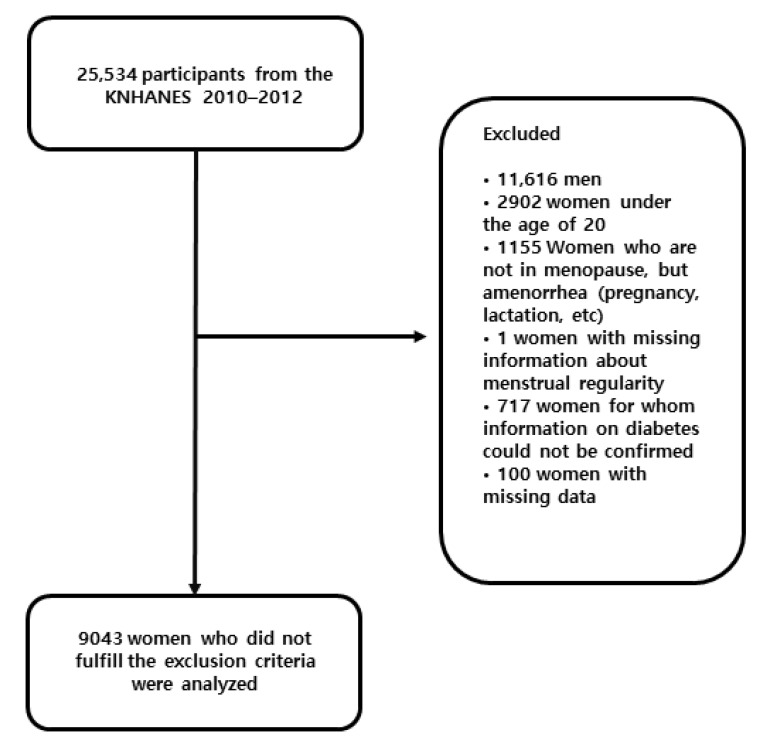
Participant flow diagram for final analysis.

**Table 1 healthcare-10-00649-t001:** Characteristics of the study participants by glucose status.

		Total (n = 9043)	Normal (n = 6796)	Impaired Fasting Glucose (n = 1436)	Diabetes (n = 811)	*p* Value
Age (years), mean (SD)		46.6 ± 15.8	**43.7 ± 15.1**	**54.6 ± 13.7**	**60.7 ± 13.2**	**<0.001**
Waist circumference		78 ± 10.1	**76 ± 9.2**	**83.7 ± 9.4**	**87 ± 10.6**	**<0.001**
BMI		23.3 ± 3.6	**22.7 ± 3.3**	**25.2 ± 3.6**	**25.8 ± 4.4**	**<0.001**
BMI (categorical)						
	BMI < 18.5	502 (6.6)	**469 (8.0)**	**18 (1.5)**	**15 (1.8)**	**<0.001**
	18.5 ≤ BMI < 25.0	5872 (65.1)	**4781 (69.9)**	**729 (50.3)**	**362 (43.3)**	
	25.0 ≤ BMI < 30.0	2277 (23.6)	**1362 (19.0)**	**577 (39.3)**	**338 (41.0)**	
	30.0 ≤ BMI	392 (4.7)	**184 (3.0)**	**112 (8.9)**	**96 (13.9)**	
Smoking						
	nonsmoker	8561 (92.9)	6421 (92.6)	1372 (94.3)	768 (94.2)	0.107
	smoker	482 (7.1)	375 (7.4)	64 (5.7)	43 (5.8)	
Hypertension status						
	normal	4470 (55.5)	**3935 (63.7)**	**416 (31.1)**	**119 (17.9)**	**<0.001**
	prehypertension	1864 (20.0)	**1369 (18.9)**	**338 (25.1)**	**157 (21.5)**	
	hypertension	2709 (24.5)	**1492 (17.3)**	**682 (43.8)**	**535 (60.6)**	
Glucose		95.1 ± 20.1	**88.4 ± 5.9**	**106.8 ± 6.4**	**141.5 ± 45.2**	**<0.001**
HbA1C		5.7 ± 0.9	**5.5 ± 0.3**	**5.9 ± 0.5**	**7.5 ± 1.5**	**<0.001**
Exercise, n (%)						
	muscle and aerobic	746 (8.4)	**581 (8.6)**	**121 (9.0)**	**44 (4.9)**	**0.027**
	aerobic exercise only	3265 (36.3)	**2461 (36.6)**	**516 (35.5)**	**288 (34.5)**	
	muscle exercise only	454 (4.7)	**351 (4.8)**	**69 (4.4)**	**34 (3.4)**	
	no exercise	4578 (50.7)	**3403 (50.0)**	**730 (51.2)**	**445 (57.2)**	
Menopausal state						
	premenopausal	4405 (59.3)	6039 (87.3)	1285 (88.7)	716 (86.4)	0.421
	postmenopausal	4638 (40.7)	757 (12.7)	151 (11.3)	95 (13.6)	
Menstrual regularity						
	regular	8040 (87.5)	**3872 (67.2)**	**419 (37.0)**	**114 (21.1)**	**<0.001**
	irregular	1003 (12.5)	**2924 (32.8)**	**1017 (63.0)**	**697 (78.9)**	
Number of pregnancies		3.2 ± 2.5	**2.9 ± 2.4**	**4.1 ± 2.3**	**4.8 ± 2.7**	**<0.001**
Oral contraceptive use						
	no	7622 (86.0)	**5843 (87.3)**	**1165 (82.8)**	**614 (78.9)**	**<0.001**
	yes	1421 (14.0)	**953 (12.7)**	**271 (17.2)**	**197 (21.1)**	

Note: Values are presented as weighted means ± standard deviations for continuous variables and unweighted sample size n (with weighted %) for categorical variables; BMI, body mass index; n, number of participants; %, percentage. The bold values indicate significant differences (*p* < 0.05).

**Table 2 healthcare-10-00649-t002:** Characteristics of study participants by glucose status in each menopausal subgroup.

		Premenopausal					Postmenopausal		
		Normal (n = 3872)	Impaired Fasting Glucose (n = 419)	Diabetes (n = 114)	*p* Value	Normal (n = 2924)	Impaired Fasting Glucose (n = 1017)	Diabetes (n = 697)	*p* Value
Age (years), mean (SD)		**60.9 ± 9.5**	**62.9 ± 9.2**	**65.7 ± 9.3**	**<0.001**	**35.3 ± 8.9**	**40.6 ± 7.5**	**42 ± 7.6**	**<0.001**
Waist circumference		**80.3 ± 8.6**	**84.7 ± 9**	**86.8 ± 10.5**	**<0.001**	**73.9 ± 8.8**	**82 ± 9.8**	**87.7 ± 11**	**<0.001**
BMI		**23.7 ± 3**	**25.1 ± 3.4**	**25.5 ± 4.4**	**<0.001**	**22.2 ± 3.4**	**25.2 ± 3.8**	**26.9 ± 4.3**	**<0.001**
BMI (categorical)									
	BMI < 18.5	**394 (2.3)**	**7 (1.1)**	**2 (1.9)**	**<0.001**	**75 (10.9)**	**11 (2.2)**	**13 (1.5)**	**<0.001**
	18.5 ≤ BMI < 25.0	**2810 (67.4)**	**216 (51.1)**	**40 (45.6)**		**1971 (71.2)**	**513 (49.0)**	**322 (34.6)**	
	25.0 ≤ BMI < 30.0	**566 (27.4)**	**147 (40.4)**	**48 (41.1)**		**796 (14.9)**	**430 (37.3)**	**290 (40.6)**	
	30.0 ≤ BMI	**102 (2.9)**	**49 (7.4)**	**24 (11.4)**		**82 (3.1)**	**63 (11.4)**	**72 (23.3)**	
Smoking status									
	nonsmoker	3599 (95.2)	397 (94.5)	106 (94.1)	0.647	2822 (91.3)	975 (93.9)	662 (94.7)	0.18
	smoker	273 (4.8)	22 (5.5)	8 (5.9)		102 (8.7)	42 (6.1)	35 (5.3)	
Hypertension status									
	normal	**2989 (35.3)**	**227 (18.9)**	**49 (11.3)**	**<0.001**	**946 (77.6)**	**189 (51.8)**	**70 (42.7)**	**<0.001**
	prehypertension	**642 (24.1)**	**118 (22.1)**	**35 (19.3)**		**727 (16.4)**	**220 (30.1)**	**122 (29.8)**	
	hypertension	**241 (40.6)**	**74 (58.9)**	**30 (69.4)**		**1251 (6.0)**	**608 (18.1)**	**505 (27.6)**	
Glucose		**89.9 ± 5.6**	**107.3 ± 6.6**	**137 ± 40.9**	**<0.001**	**87.7 ± 5.8**	**105.8 ± 6**	**158.3 ± 55.4**	**<0.001**
HbA1C		**5.6 ± 0.3**	**6 ± 0.5**	**7.4 ± 1.4**	**<0.001**	**5.4 ± 0.3**	**5.7 ± 0.4**	**7.8 ± 1.9**	**<0.001**
Exercise, n (%)									
	muscle and aerobic	331 (8.7)	48 (7.3)	6 (4.7)	0.017	250 (8.6)	73 (11.9)	38 (5.7)	0.455
	aerobic exercise only	1431 (34.9)	152 (35.7)	39 (34.8)		1030 (37.4)	364 (35.0)	249 (33.3)	
	muscle exercise only	181 (5.5)	20 (4.5)	7 (3.2)		170 (4.5)	49 (4.2)	27 (4.2)	
	no exercise	1929 (50.8)	199 (52.5)	62 (57.3)		1474 (49.5)	531 (48.9)	383 (56.8)	
Menstrual regularity									
	regular	3344 (91.7)	363 (89.9)	87 (88.7)	0.122	2695 (85.2)	922 (86.7)	629 (77.6)	0.135
	irregular	528 (8.3)	56 (10.1)	27 (11.3)		229 (14.8)	95 (13.3)	68 (22.4)	
Number of pregnancies		**4.6 ± 2.3**	**4.8 ± 2.3**	**5.3 ± 2.7**	**<0.001**	**2 ± 1.9**	**2.8 ± 1.7**	**3.1 ± 2**	**<0.001**
Oral contraceptive use									
	no	3556 (79.5)	382 (78.2)	108 (75.4)	0.144	2287 (91.1)	783 (90.5)	506 (91.9)	0.915
	yes	316 (20.5)	37 (21.8)	6 (24.6)		637 (8.9)	234 (9.5)	191 (8.1)	

Note: Values are presented as weighted means ± standard deviations for continuous variables and unweighted sample size n (with weighted %) for categorical variables; BMI, body mass index; n, number of participants; %, percentage. The bold values indicate significant differences (*p* < 0.05).

**Table 3 healthcare-10-00649-t003:** Multivariate logistic regression analysis results of risk for impaired fasting glucose and diabetes according to menopausal status and menstrual irregularities.

			Risk for Impaired Fasting Glucose Compared to Normal			Risk for Diabetes Compared to Normal		
			OR	95% CI	*p* Value	OR	95% CI	*p* Value
Total	Menopausal state							
		premenopausal	1.00	reference		1.00	reference	
		postmenopausal	1.16 ^‡^	(0.88, 1.51)	0.286	1.51 ^‡^	(1.01, 2.27)	0.047
	Menstrual regularity							
		regular	1.00	reference		1.00	reference	
		irregular	0.99 ^‡^	(0.76, 1.28)	0.916	1.51 ^‡^	(1.1, 2.07)	0.011
Premenopausal	Menstrual regularity							
		regular	1.00	reference		1.00	reference	
		irregular	0.68 ^‡^	(0.46, 1.02)	0.06	1.22 ^‡^	(0.64, 2.32)	0.555
Postmenopausal	Menstrual regularity							
		regular	1.00	reference		1.00	reference	
		irregular	1.32 ^‡^	(0.94, 1.84)	0.11	1.65 ^‡^	(1.12, 2.42)	0.012

^‡^: Adjusted for number of pregnancies, oral contraceptive pill use, age, BMI, smoking status, blood pressure status, and type of exercise.

## Data Availability

The data evaluated in this study cannot be uploaded publicly due to legal restrictions, concerns for patient privacy, and third-party ownership of the data by the Korea National Health & Nutrition Examination Survey (KNHANES). However, the data are directly obtainable upon request, either by accessing https://knhanes.cdc.go.kr/knhanes/eng/index.do (accessed on 8 December 2021) or by emailing knhanes@korea.kr.

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
