# Peer review of "The Association between Menstrual Irregularities and the Risk of Diabetes in Premenopausal and Postmenopausal Women: A Cross-Sectional Study of a Nationally Representative Sample"

_healthcare, 2022, doi:10.3390/healthcare10040649_

Round 1
Reviewer 1 Report
The paper concludes; 'This study emphasizes the necessity of monitoring and early prevention, along with medical advice on 30 menstrual irregularities, to reduce the prevalence of diabetes and improve the quality of life of post- 31 menopausal women.'. However, its already known that menstrual irregularity is mainly depend on anovulatory cycles both in early puberty and perimenopausal period, and the authors have tried to relate their findings to PCOS. The higher rate of menstrual irregularities can be due to metabolic changes due to the presence of diabetes, thats what if you treat diabetes you'll also treat menstrual irregularities. Otherwise you can not prevent diabetes by commencing progesterone or etc. to treat any menstrual irregularitiy.
Reviewer 2 Report
Summary: Authors studied association between effects of menstrual irregularities and menopause on the risk of diabetes mellitus. They concluded that menstrual irregularities were a risk factor for postmenopausal diabetes, menopausal status appeared to independently affected diabetes risk.
Major comments:
- If the article is focused on menstrual irregularities, menopausal women should be excluded - this approach could be important specifically for premenopausal and not menopausal women. Therefore, the first sentence in Conclusion (rr. 270-272) does not match the main aim of the study.
- Better definition of menopause (STRAW criteria: Harlow, S.D., et al. STRAW+10 Collaborative Group. Executive summary of the Stages of Reproductive Aging Workshop +10: addressing the unfinished agenda of staging reproductive aging. Climacteric. 2012. 15:105-114. …) and menopausal irregularities could be presented including more detailed description of hormone (substitutional/replacement?) therapy and/or type of contraception.
Minor comments:
- Pls look at these papers which assessed menstrual irregularities: Khoudary SR et al. Patterns of menstrual cycle length over the menopause transition are associated with subclinical atherosclerosis after menopause. Menopause. 2021;29(1):8-15. doi: 10.1097/GME.0000000000001876. PMID: 34636354. Santoro N, et al. Menstrual Cycle Hormone Changes in Women Traversing Menopause: Study of Women's Health Across the Nation. J Clin Endocrinol Metab. 2017;102(7):2218-2229. doi: 10.1210/jc.2016-4017. PMID: 28368525; PMCID: PMC5505186.
- Diabetes mellitus should be defined - suppose type 2
- Terms „svyglm, svychisq“ (rr.140-144) should be explained.
Conclusion: If the main aim of study is focused on menstrual irregularities and risk of diabetes mellitus, menopausal women should be excluded. Other comments, pls see above.
Reviewer 3 Report
The paper entitled "The association between menstrual irregularities and the risk 2 of diabetes in premenopausal and postmenopausal women: A 3 cross-sectional study of a nationally representative sample" examined the association between menstrual irregularities and menopause and risk for developing type 2 diabetes
The paper is well written, the information from a statistical point of view and observation may be interesting, but there is a lack of scientific soundness as far as pathophysiology of type 2 diabetes and true risk factors are concerned, such as genetic predisposition, insulin resistance, hyperinsulinemia and environmental factors, such as diet, dietary habits and exercise levels.
The manuscript would benefit if authors discussed the risk for developing type 2 diabetes based on genetic predisposition and hyperinsulinemia/insulin resistance which are the underlying and determinant factors for developing type 2 diabetes in the introduction and discussion. Please read the following papers:
Mitrou P et al. 2009;94(8):2958-61 doi: 10.1210/jc.2008-2297 J Clin Endocrinol Metab &
Dimitriadis, G. D., et al. (2021). "Regulation of Postabsorptive and Postprandial Glucose Metabolism by Insulin-Dependent and Insulin-Independent Mechanisms: An Integrative Approach." Nutrients 13(1).
It is evident that not all pre- or post-menopausal women will develop type 2 diabetes, regardless of menstrual irregularities. It is also evident that even not all obese women will develop type 2 diabetes, regardless of menstrual irregularities. Those however, who are pre-post menopausal or menstrual irregularities or PCOS and have hyperinsulinemia or insulin resistance or both or had gestational diabetes are at a high risk for developing type 2 diabetes. Thus, menstrual irregularities or menopause by themselves are poor risk factors for developing type 2 diabetes if a woman does not have a genetic predisposition or hyperinsulinemia/insulin resistance (in the case that she is not aware of genetic predisposition, i.e., she has no information about health status of her relatives for many reasons). It would be interesting if you had data on gestational diabetes in these women and you should add the data. Also, PCOS is a significant risk factor for type 2 diabetes due to the hyperinsulinemia and insulin resistance significant characteristics of this disease state and not PCOS by itself as a health condition. Do the authors have data on fasting insulin and thus, HOMA calculation? If yes, it should be added and discussed. Moreover, the authors have not discussed diet and dietary habits issues which are very important to type 2 diabetes development. For example, how did these women gain weight? high fat diet, low exercise levels, high carbohydrate diet, eating late at night, having irregular meals? All of the above? If the authors have information about these they should be added as they are far more important than just menstrual irregularities or menopause. Finally, even if mestrual irregularities by itself increase the risk for developing type 2 diabetes what is the mechanism, how do they interfere with glucose metabolism? insulin receptors? insulin secretion?
Reviewer 4 Report
I rate the manuscript very highly. I read the manuscript with interest, the topic is relevant, the study protocol is well planned. Study described properly. Conclusions supported by the results.
I only have the following questions:
1. Table 1 shows the number of smokers and non-smokers. Which category includes ex-smokers? Information on the degree of exposure is missing.
2. The percentage of patients with arterial hypertension is indicated. Information on sBP and dBP values is missing, as well as consideration of the importance of antihypertensive treatment.
3. I would suggest taking into account the problem of insulin resistance, for example by indicating HOMA-IR.
Reviewer 5 Report
The authors studied the associations between menstrual irregularities and the risk of diabetes in pre- and postmenopausal women. The manuscript is interested, but some changes should be checked to consider this article suitable for publication in this journal.
- The introduction is of short extension. The role of menstrual irregularities and the risk of diabetes have been widely discussed separately in many settings and should be commented to introduce the topic. It would be helpful to understand the manuscript better.
- The section "results" seems repetitive in almost all paragraphs. Line 198 "this study reported...", line 206 "in our study", line 224 "the data in this study", line 227 "in this study", please rewrite. Overall, the words "study" or "studies" are used repetitively thoughout the text not only to refer this manuscript, but also to other investigations, which becomes redundant. The authors should use synonyms to improve it.
- The fact that this is the first study to examine the relationship between menstrual irregularities and diabetes before and after menopause is repeated twice (lines 203 and 252). Only once is enough in the text. Also, that sentence in the abstract could atract the attention from the readers.
- There are some spelling mistakes. For example, in line 244 it should be "suggests". Please revise the manuscript.
- The conclusion is very poor taking into account the novelty of this study (the first time that menstrual irregularities and diabetes before and after menopause have found to be associated). The authors should give their opinion about the implications of this relationship and the new avenues of research that could be carried out. Are necessary further research to elucidate the associations? What kind of studies? How could the limitations be improved?
Reviewer 6 Report
Line 90, Is this based on international guidelines for the definition of menstrual irregularities " (missing three or more periods in a row)? It may be considered too long, and there may be a risk that only extreme cases are included. Line 185, Any guesses as to why the risk of diabetes remained the same for premenopausal subjects with and without menstrual irregularities? To be included in the text.Author Response
Please see the attachment

Round 2
Reviewer 1 Report
The explanation by the authors are not adequate and the hypothesis should be confirmed with prospective data.
Author Response
We appreciate the time and effort you have dedicated to providing insightful feedback. Thank you for your valuable comments. For follow-up studies, prospective follow-up studies and mechanistic studies are needed. This content is attached to the conclusion (line 295-302).
line 295-302: However, since this is a cross-sectional study involving a survey, future studies are needed to confirm our observations and/or extend our findings. Well-designed prospective long-term follow-up studies or mechanistic studies are needed to clarify the causal relationship, and based on their findings, stronger suggestions and guidelines can be presented. Nevertheless, our findings support the need for ongoing diabetes-related monitoring and screening among women with a history of menstrual irregularities along with medical advice on menstrual irregularities to reduce the prevalence of diabetes and improve the quality of life of postmenopausal women.
Reviewer 2 Report
Corrections made by authors are in general acceptable. Neverthelless I strongly recommend to strictly differentiate between premenopausal and menopausal women especially regarding irregularity of menstrual bleeding i.e. menopause itself as a risk factor x bleeding irreguralities before menopause but in menopausal women. For example, in Table 1 menstrual irregularities and menopausal status are mixed - in other words in women after menopause the bleeding irregularities should be at least presented/described separately from premenopausal women; optimally analyzed separately.
Author Response
We appreciate the time and effort you have dedicated to providing insightful feedback. We agree with your comments. That point was added to the limitations in the discussion (line 276-278)
Revision : In other words, it is not clear whether the cause of irregular menstrual cycles for women in their 40’s is related to menopausal transition or an enduring irregular menstrual pattern.
Reviewer 4 Report
The authors responded to my comments. Some changes have been made to the manuscript in line with my suggestions. The remaining issues were considered as limitations of the study - I accept this solution.
Author Response
We appreciate the time and effort you have dedicated to providing insightful feedback.
Reviewer 5 Report
I thank the authors their time to improve the manuscript and their consideration with my suggestions. Now it can be accepted by the journal.
Author Response

(The authors gave the same response as above.)

Reviewer 6 Report
none
Author Response

(The authors gave the same response as above.)
